# Physicochemical Characterization and Antioxidant Activity of Jara Honey Produced in Western Georgia

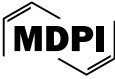

Nona Abashidze *, Indira Djafaridze *, Maia Vanidze, Meri Khakhutaishvili, Maia Kharadze, Inga Kartsivadze, Ruslan Davitadze and Aleko Kalandia

Department of Chemistry, Faculty of Natural Sciences and Health Care, Batumi Shota Rustaveli State University (BSU), 6010 Batumi, Georgia; maia.vanidze@bsu.edu.ge (M.V.); meri.khakhutaishvili@bsu.edu.ge (M.K.); maia.kharadze@bsu.edu.ge (M.K.); inga.qarcivadze@bsu.edu.ge (I.K.); ruslan.davitadze@bsu.edu.ge (R.D.); aleko.kalandia@bsu.edu.ge (A.K.)
* Correspondence: nona.abashidze@bsu.edu.ge (N.A.); indira.jafaridze@bsu.edu.ge (I.D.)

**Abstract:** The purpose of this research article was to study the physicochemical characteristics of semi-wild Jara honey grown in Western Georgia. Jara honey is produced in the alpine and sub-alpine forest zone of high mountain Adjara, which is distinguished by its variety of honey plants. The physicochemical characteristics were examined concerning the Alimemtarius Code and EU regulations: moisture content, total carbohydrates, free acidity, pH, electrical conductivity, microelements (Li, Na, K, Mg, Ca), color, total phenols, total phenolic acids, total flavonoids, proline, diastase activity, proteins, and microscopic study of pollens. Using the UPLC-MSB method, grayanotoxin-III was identified in the semi-wild Jara honey samples. The findings demonstrated that the honey has significant concentrations of phenols, phenolic acids, and flavonoids. A directly proportional relationship was established between the quantitative content of phenolic compounds and the antioxidant activity of honey. This article is the first study of the characteristics of Jara honey produced in Western Georgia.

**Keywords:** honey; antioxidant activity; physicochemical; melissopalynology; grayanotoxin

## 1. Introduction

Honey is the most recognized and famous natural food produced by bees (*Apis mellifera*) from nectar and honeydew [1]. Its historical, cultural, and economic significance makes it the major beekeeping product [2]. Honey is an aromatic, sweet natural product with high nutritional value that has been used by humans since time immemorial [3].

Due to its taste and nutritional value, honey has been commonly consumed as a food for thousands of years [4]. It is used as a component in the food industry [5] and also serves as a food preservative [4,6]. Moreover, honey is also used in medicine [7] and in the production of cosmetics [8] due to the numerous positive properties that can be attributed to the various bioactive molecules it contains [2].

Honey has a very complex composition [1]. It contains more than 200 compounds [5]. It is mainly composed of sugars (70–80%) [9], namely 38% fructose, 31% glucose, 10% other sugar types [10], and water (10–20%) [10]. The uniqueness of this food product resides in the rich spectrum of compounds that do not make up the bulk of the mass. For instance, honey also contains proteins, organic acids (such as acetic acid and gluconic acid, etc.) [5], lipids, carotenoids, minerals [11], and enzymes [9,12]. More importantly, bioactive components that are crucial in determining the specific and individual characteristics of honey, such as flavonoids, phenolic acids, and vitamins (ascorbic acid, niacin, etc.), are also present in minor quantities [11,13–15]. Honey can originate from single or multiple plant species [16]. Accordingly, many different types of honey are available on the global market. Following its origin, honey is divided into blossom, honeydew, monofloral, and multifloral varieties based on the source of the honey [16–19].

Although Georgia is a small country, it is a very important region for the development of beekeeping, where it is possible to produce high-quality honey of various origins [20]. In this regard, the uniqueness of Georgia is determined by its geographical location, climatic conditions, and rich and varied vegetation [21,22]. Georgia mainly produces five types of honey in large quantities. These are acacia honey, blossom honey, alpine honey, linden honey, and chestnut honey [23]. In Georgia, we also have traditional Georgian semi-wild Jara beekeeping. Jara honey is produced in the alpine and sub-alpine forest zones of the high mountains of Adjara, which are distinguished by the variety of honey plants [21]. A Jara is a hollowed log cut in two. It is mainly carved from the linden tree (*Tilia begoniifolia Steven*), which is chosen for its light weight and lack of specific smell to avoid disturbing the bees [15,21,24]. The Jara is positioned on trees in the forest, and beekeepers no longer interfere in the development of the bee colony, allowing for the completely natural and unmanaged production of Jara honey. There is also no resource management for honey bees. Pesticides and antibiotics are also not used [25]. Although beekeepers in Georgia typically use artificial hives, the bees created their hives (without human intervention) during the production of Jara honey. The obtained honey is matured in natural pine. Jara honey is collected once a year, at the beginning of autumn [21,22].

It should be mentioned that only the Adjara region of Georgia produces Jara honey. Its cost is consistently two to three times greater compared to the price of other Georgian honey. The "Kakhetian Traditional Winemaking" enterprise collects Jara honey (https://ktwshop.ge/en/27-honey, accessed on 15 July 2024).

It is also worth noting that the flowering of honey plants in the highlands of Adjara is preceded by the flowering of the rhododendron, which contains a toxin, in cold and late spring conditions. Therefore, there is an increased probability that the bee consumes the rhododendron nectar and that a toxin seeps into the content of Jara honey, which enters the human body and ultimately poisons humans [18,26–28].

Our study aimed to investigate the chemical and physical characteristics, biologically active ingredient content, and antioxidant activity of Jara honey, as an almost "unknown" product, along with determining its pollen concentration. To control uncontrolled honey, we were able to define Jara honey, as a natural and unique product, based on the data we gained.

## 2. Materials and Methods

### 2.1. Honey Samples

Honey samples were provided by beekeepers and collected from three different municipalities—Adjara (Figures 1 and 2). Honey samples were provided by private beekeepers and the company "Kakhetian Traditional Winemaking". The harvest took place in the fall of 2018 (in September) (Table 1).

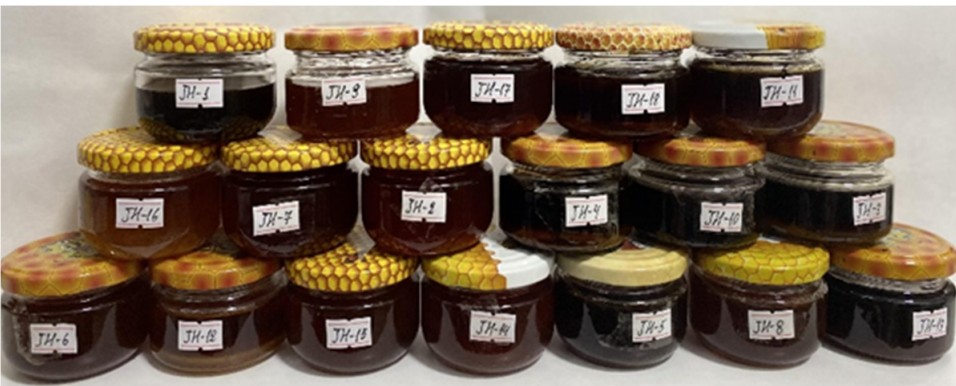

**Figure 1.** Samples of Jara honey.

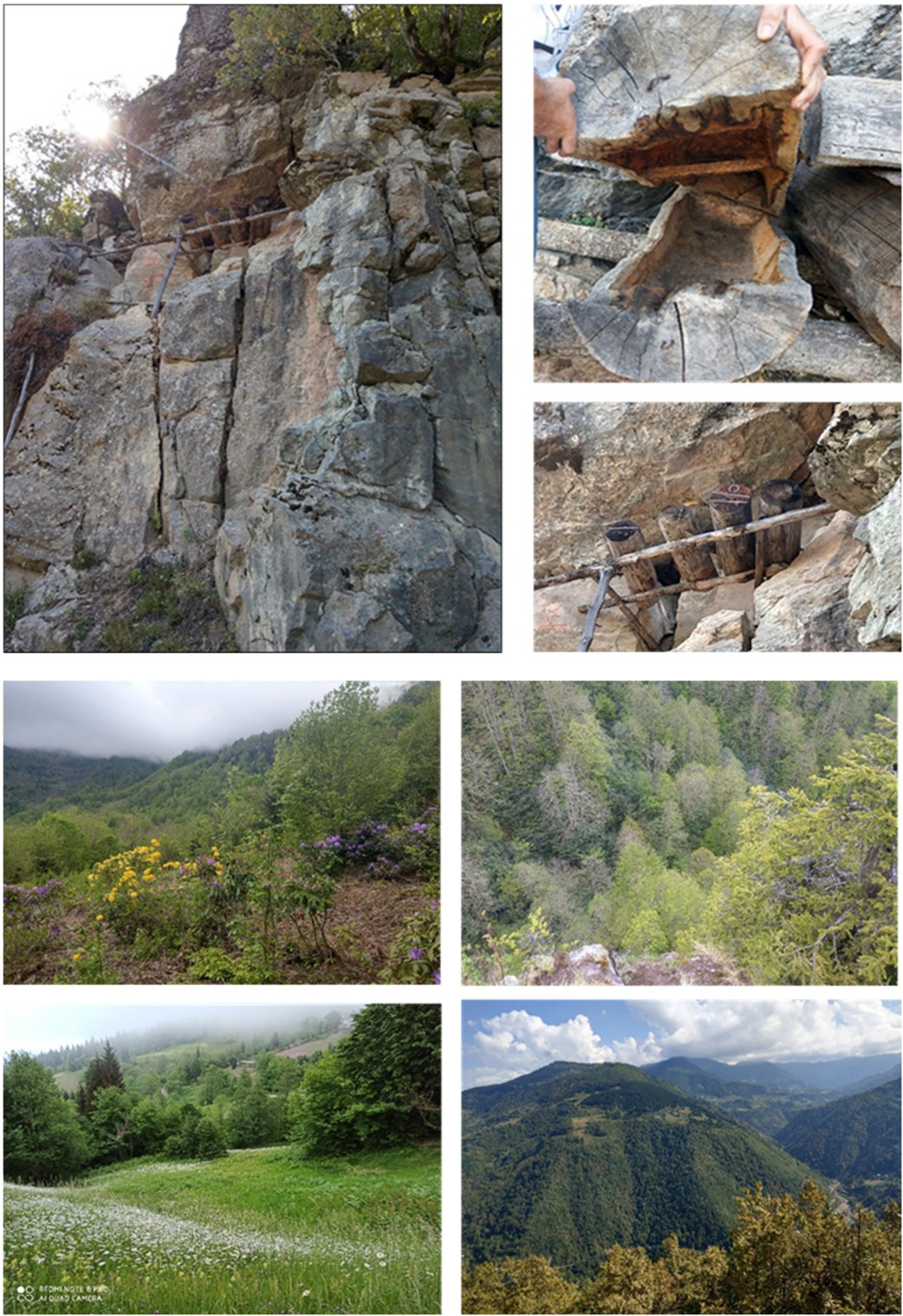

**Figure 2.** Pictures of Jara hives and the plants from which Jara honey is obtained (https://www.jarahoney.com/, accessed on 15 July 2024).

Honey samples were stored in hermetic containers (4–5 °C). In the case of impurities, the honey was passed through a grid with a diameter of 0.5 mm.

**Table 1.** Sample of Jara honey were taken for analysis.

| Samples | Samplers Code | Sampling Place (Municipality) | Height above Mean Sea Level, m | Coordinates |
|---------|---------------|-------------------------------|-------------------------------|-------------|
| Jara Honey 1 | JH-1 | Keda–Gobroneti | 607 | 41°39′8.5″ N, 42°2′18.6″ E |
| Jara Honey 2 | JH-2 | Keda–Zesopeli | 536 | 41°37′14.2″ N, 41°57′44.6″ E |
| Jara Honey 3 | JH-3 | Keda–Tskhmorisi | 523 | 41°38′22″ N, 42°2′50″ E |
| Jara Honey 4 | JH-4 | Keda–Zendidi | 336 | 41°36′10″ N, 41°55′55″ E |
| Jara Honey 5 | JH-5 | Keda–Zundaga | 400 | 41°34′42.4″ N, 41°49′32.9″ E |
| Jara Honey 6 | JH-6 | Keda–Namonastrevi | 820 | 41°34′13″ N, 42°3′10″ E |
| Jara Honey 7 | JH-7 | Keda–Silibauri | 490 | 41°34′46.3″ N, 42°0′47.1″ E |
| Jara Honey 8 | JH-8 | Keda–Medzibna | 640 | 41°33′51.9″ N, 41°58′29″ E |
| Jara Honey 9 | JH-9 | Keda–Merisi | 700 | 41°34′54″ N, 42°0′18″ E |
| Jara Honey 10 | JH-10 | Shuakhevi–Instkirveli | 1385 | 41°42′57.4″ N, 42°14′9.3″ E |
| Jara Honey 11 | JH-11 | Shuakhevi–Khabelashvilebi | 766 | 41°42′28″ N, 42°10′45.7″ E |
| Jara Honey 12 | JH-12 | Shuakhevi–Kidzinidzeebi | 1040 | 41°35′6.1″ N, 42°11′37.4″ E |
| Jara Honey 13 | JH-13 | Shuakhevi–Karapeti | 1334 | 41°33′8.1″ N, 42°15′24.5″ E |
| Jara Honey 14 | JH-14 | Khulo–Skhalta | 800 | 41°35′4.84″ N, 42°19′46.21″ E |
| Jara Honey 15 | JH-15 | Khulo–Kvatia | 1090 | 41°34′40″ N, 42°24′17″ E |
| Jara Honey 16 | JH-16 | Khulo–Pushrukauli | 1180 | 41°33′41″ N, 42°27′0″ E |
| Jara Honey 17 | JH-17 | Khulo–Rakvta | 1350 | 41°34′11″ N, 42°28′58″ E |
| Jara Honey 18 | JH-18 | Khulo–Bardnali | 570 | 42°36′22″ N, 42°42′50″ E |

## 2.2. Methods

Measurements of absorbance were performed using a UV–VIS spectrophotometer model UV-5 (Mettler Toledo, Greifensee, Switzerland). The chemicals and reagents used were of analytical grade and came from Sigma and Merck Chemical Company (Darmstadt, Germany). We used an HPLC (Water 2414 Refractive Index and Waters 2489 UV/Visible detector) (Vienna, Austria) and a UPLC with MS detector (Waters Acquity H-Class, USA), and a PDA Detector (Waters Acquity) (Vienna, Austria) for identification of compounds, as well as a UKA Technic professional light microscope (Casper, WY, USA) for melissopalynological analysis.

## 2.3. Determination of Moisture Content and Total Soluble Solids (% Brix)

The Refractometric method was used to determine the moisture, or conversely, the soluble solids in the honey, as previously described [29]. This was accomplished by measuring the refractive index of the honey using a refractometer thermostated at 20 °C and regularly calibrated with distilled water. This method is based on the principle that the refractive index increases with the solid content [3]. The homogenized sample was put in a

flask and placed in a water bath at 50 °C (±0.2) until all the sugar crystals were dissolved. The resulting solution was cooled to room temperature, stirred, and immediately placed evenly on the surface of the refractometer prism. Each sample was measured three times and the average value was taken. Moisture content and soluble solids were obtained from the refractive index of the honey by referring to a standard table [3,29].

### 2.4. Determination of pH

The pH was measured using a digital pH meter which was calibrated at room temperature using buffer solutions at pH 4 and 7. Five grams of each honey sample was diluted with 50 mL of distilled water to make a 10% solution [29,30].

### 2.5. Determination of Free Acidity

The acidity of honey was determined by the volumetric method. Ten grams of honey was dissolved in 75 mL of distilled water and the solution was titrated with 0.1 M NaOH to pH 8.30. Acidity is expressed in milliequivalents acids/kg honey (mEq/kg) [31,32].

### 2.6. Determination of Electrical Conductivity

Electrical conductivity was measured at 20 °C in solutions of honey samples (20.0 g of honey in volume solution in 100 mL distilled water) [33] using a conductometer (Mettler Toledo).

### 2.7. Determination of Diastase Activity (Standard Schade Method)

The unit of diastase activity, the Schade unit, is defined as the amount of enzymes that will convert 0.01 g of starch to the prescribed end-point in 1 h at 400 °C under defined conditions. The principle of this method is as follows: a standard solution of starch, which is capable of producing color in a defined range of intensity when treated with iodine, is acted upon by an enzyme in the sample being investigated under standard conditions. The resulting reduction in blue coloration is then measured at defined intervals. A plot of absorbance relative to time or the regression equation is used to determine the time (tx) required to reach the specified absorbance of 0.235. The diastase number is calculated as 300 divided by tx [34].

### 2.8. Determination of Proline

An amount of 5 g of honey was weighed into a beaker and dissolved in about 50 mL of distilled water. Then, the solution was transferred quantitatively to a 100 mL volumetric flask, diluted to volume with distilled water, and well shaken very well. An amount of 1 mL of the sample solution was put in each of the two tubes, and 0.5 mL of formic acid (98–100%) and 2 mL of ninhydrin solution (3% in ethylene glycol monomethyl ether) were added to each tube. The tubes were capped carefully and then they were shaken vigorously. In the same way, 1 mL of distilled water was added to one tube instead of being added to the sample solution and then the previous procedure was followed. In both cases: The tubes were placed in a boiling water bath for 15 min and were transferred to a water bath of 22 °C for 10 min; 10 mL of the 2-propanol–water solution (1:1) was added to each tube in regular intervals of time. The tubes cooled at 22 °C were removed before 35 min and the absorbances at 520 nm determined. The strict control of time in each step is critical [33,34].

### 2.9. Determination of Color Intensity

Color intensity was determined according to Ferreira et al. and Lacerda et al. [35]. Honey samples were diluted to 50% with distilled water, mixed, and centrifuged at 3200 rpm/5 min. The absorbance was measured at 635 nm using a spectrophotometer, and the color intensity was determined using the Pfund scale using the following equation (Pfund = −38.70 + 371.39 Abs) [35,36].

### 2.10. Determination of Total Phenolic Content

An amount of 500 μL of honey solution was added to 2.5 mL of Folin Ciocalteu (0.2 N). After 5 min, 2 mL of sodium carbonate solution (75 g/L) was added and the solution was incubated for 2 h in the dark. The absorbance was measured at 760 nm in a spectrophotometer [37,38]. The standard curve was defined by known concentrations of gallic acid, ranging between 0 and 200 mg L1. The results were expressed in milligrams for gallic acid equivalents (CE)/kg of honey.

### 2.11. Determination of Total Flavonoids Content

The honey solution was prepared with methanol 50% and was previously homogenized and filtered through a quantitative filter; then, 1 mL of a honey solution (1 mg/mL) was mixed with 0.3 mL $NaNO_2$ (5%), and a solution of 0.3 mL $AlCl_3$ (10%) was added in five minutes. The honey samples were mixed and in six minutes they were neutralized with 2 mL of NaOH solution (1 M). The absorbance was measured for all samples at 510 nm. The results were expressed in mg for quercetin equivalents (CE)/kg of honey [39,40].

### 2.12. Determination of Phenolic Acids

The total content was determined according to the method of Mazza, Fukumoto, Delaquis, Girard, and Ewert. Appropriate aliquots of solutions prepared with honey, acidified ethanol solution (250 μL) (0.1% HCl in 95% ethanol), and 2%HCl (4.55 mL) were transferred to a 10 mL volumetric flask and incubated for 15 min, and then the absorbance was determined at 320 nm. The standard curve was defined by caffeic acid (0–0.8 μg mL$^{-1}$). The results were expressed in milligrams for caffeic acid equivalents kg of honey [4,41].

### 2.13. Determination of Antioxidant Activity (Assay with DPPH)

The DPPH assay was performed by using 750 μL of honey solution that was mixed with 1.5 mL of DPPH solution in methanol (0.02 mg mL$^{-1}$). The mixture was homogenized for 30 min at room temperature and then the absorbance was determined at 517 nm [3,39]. The antioxidant activity was calculated as the 50% inhibition of 0.1 mM DPPH radical by mg of honey sample.

### 2.14. Determination of Carbohydrates

The honey solution (5%) was prepared with methanol (25% solution). After filtration of the solution, the sugar content was determined by Waters HPLC (High Pressure Liquid Chromatography) with RI detection. In the mobile phase, the ratio of acetonitrile to water was as follows: 80:20, *v/v*. Peaks were identified based on their retention times. Quantitation was performed according to the external standard method on peak areas or peak heights [34,35].

### 2.15. Determination of Cations

The honey solution (5%) was prepared with water. After filtration of the solution, the minerals were determined by the Waters 1515 isocratic HPLC Pump (High-Pressure Liquid Chromatography) with Waters 432 Conductivity detection (Vienna, Austria) with the following specifications: Column-IC-Pak C M/D; Mobile phase: 0.1 mM EDTA/3 mM $HNO_3$. Peaks were identified based on their retention times. Quantitation was performed according to the external standard method on peak areas or peak heights [32,42].

### 2.16. Determination of Protein

Determining the level of protein in the honey was based on the modification of the nitrogen of the sample into ammonium sulfate through acid digestion, distillation, and the subsequent release of ammonia, which is mixed in an acidic solution and titrated. Determining the nitrogen and the conversion factor provided the crude protein result, based on the Kjeldahl method [16,42].

### 2.17. Determination of Pollen Analysis

Ten grams of each honey was dissolved in 20 mL of warm water (40 °C). The solution was centrifuged for 5 min at 4000 r/min, the supernatant solution was decanted, and the centrifugal step was repeated twice to remove excess water. The sediments were blended with glycerin. Two slides were prepared from each sample and photographed under a light microscope [10]. The identification of the pollen grains was based on their size and shape [31,32,43].

### 2.18. Determination of Grayanotoxin-III Analysis

Approximately 5 g of the honey sample was extracted with 30 mL methanol in a flask attached to a condenser at 60 °C for 6 h. The extract was subsequently filtered to remove particles, and the final volume was determined, with 5 mL methanol extract being set aside for antioxidant activity analyses. The remaining extract was evaporated until dry using a rotary evaporator (INGOS RVO 400) at 40 °C. The residues were dissolved in 10 mL distilled water and transferred to a C18 solid phase extraction (SPE) cartridge (Waters Sep-Pak, Vac 6cc, C18–500 mg) which was initially conditioned with 5 mL methanol followed by 5 mL water. The cartridge was washed with 5 mL water to remove unbound materials. GTX-III was eluted from C18 SPE using 5 mL methanol. Finally, the organic solvents were evaporated in a rotary evaporator under reduced pressure at 40 °C. The residue was weighed and dissolved in methanol for LC/MS-MS analysis [26,28,44]. Liquid chromatography–tandem mass spectrometry (Waters, UPLC Acquity, QDa Detector) was used for the identification of grayanotoxin-III. The analytical column was an Acquity UPLC BEN C18. GTX-III was eluted using a mobile phase consisting of a 50:50 water/methanol solution containing 1% acetic acid at a flow rate of 0.3 mL/min in 8 min. Limits of detection and quantification were as follows: LOD—0.010 mg/L and LOQ—0.035 mg/L; collision energy—30 eV [28].

### 2.19. Statistical Analysis

All experimental assays were performed in triplicate. The results obtained were expressed as mean values ± standard deviation (SD). Excel software (Office 2019) was used to calculate each dataset's standard error. $p \leq 0.05$ was the confidence coefficient [45]. The correlation relationship was defined using "Data Analysis" (Analysis Tools–Correlation).

## 3. Results and Discussion

### 3.1. Moisture Content (%)

One of honey's most crucial properties is its water content because it affects the substance's viscosity, specific gravity, maturity, crystallization, flavor, preservation, shelf life, and palatability [44]. It is a crucial element in determining the quality of honey [46]. It depends on several variables, including bee species, floral supply, honey harvesting season, honey maturation level (total dehydration), and meteorological factors [16,27].

Excessive water content in honey can lead to the growth of microbiological reactions and the development of the fermentation process, which negatively affects the qualitative characteristics of honey [3]. The moisture level in this inquiry ranged from 14.3% to 18.6%. (Table 2). Specifically, the average rate in the Keda samples was 17.78%, in the samples of Shuakhevi it was 16.47%, and in the samples of Khulo it was 15.9%. The water content in the honey samples taken for analysis was less than 20%, which is within the norm stipulated by the standard (International Honey Commission). This also indicated the maturity of the analyzed honey samples.

**Table 2.** Jara honey characteristics of water, dry substances, free acids, pH, and electrical conductivity in Western Georgia.

| Name | Moisture (%) | Dry Substances, g/100 g | Free Acidity, mEq/kg | pH | Electrical Conductivity (μs/sm) |
|---|---|---|---|---|---|
| JH-1 | 17.0 ± 0.26 | 83.0 ± 1.25 | 22.48 ± 34 | 5.18 ± 0.08 | 1.185 ± 0.018 |
| JH-2 | 17.8 ± 0.26 | 82.2 ± 1.23 | 26.4 ± 0.40 | 5.3 ± 0.08 | 1.159 ± 0.017 |
| JH-3 | 17.0 ± 0.26 | 83.0 ± 1.25 | 27.5 ± 0.41 | 4.23 ± 0.06 | 1.071 ± 0.016 |
| JH-4 | 17.36 ± 0.26 | 82.6 ± 1.24 | 23.32 ± 0.35 | 5.14 ± 0.08 | 1.099 ± 0.016 |
| JH-5 | 18.4 ± 0.27 | 81.6 ± 1.22 | 25.04 ± 0.38 | 4.86 ± 0.07 | 1.706 ± 0.026 |
| JH-6 | 18.2 ± 0.27 | 81.8 ± 1.23 | 23.08 ± 0.35 | 5.06 ± 0.08 | 1.552 ± 0.023 |
| JH-7 | 17.6 ± 0.26 | 82.4 ± 1.24 | 27.00 ± 0.41 | 5.2 ± 0.08 | 1.606 ± 0.024 |
| JH-8 | 18.1 ± 0.27 | 81.9 ± 1.23 | 20.96 ± 0.31 | 5.62 ± 0.08 | 1.181 ± 0.018 |
| JH-9 | 18.6 ± 0.27 | 81.4 ± 1.22 | 23.24 ± 0.41 | 5.63 ± 0.08 | 1.447 ± 0.022 |
| JH-10 | 18.18 ± 0.27 | 81.8 ± 1.23 | 28.5 ± 0.43 | 5.07 ± 0.08 | 1.111 ± 0.017 |
| JH-11 | 16.1 ± 0.24 | 83.9 ± 1.26 | 23.28 ± 0.35 | 5.11 ± 0.08 | 1.1742 ± 0.018 |
| JH-12 | 15.0 ± 0.23 | 85.0 ± 1.28 | 27.5 ± 0.41 | 4.71 ± 0.07 | 1.1008 ± 0.017 |
| JH-13 | 16.6 ± 0.24 | 83.4 ± 1.25 | 24.96 ± 0.37 | 4.96 ± 0.07 | 1.190 ± 0.020 |
| JH-14 | 14.3 ± 0.23 | 85.7 ± 1.29 | 25.3 ± 0.38 | 5.09 ± 0.08 | 1.122 ± 0.017 |
| JH-15 | 15.4 ± 0.23 | 84.6 ± 1.27 | 28.41 ± 0.43 | 5.31 ± 0.08 | 1.166 ± 0.017 |
| JH-16 | 16.4 ± 0.24 | 83.6 ± 1.25 | 25.64 ± 0.38 | 5.17 ± 0.08 | 1.150 ± 0.017 |
| JH-17 | 18.4 ± 0.27 | 84.6 ± 1.27 | 24.4 ± 0.37 | 5.14 ± 0.08 | 1.137 ± 0.017 |
| JH-18 | 18.0 ± 0.27 | 82.0 ± 1.23 | 26.64 ± 0.40 | 4.4 ± 0.07 | 1.123 ± 0.017 |

*3.2. Sugars*

Sugar is a crucial component of honey [47], primarily existing in the form of monosaccharides and disaccharides. The total content of monosaccharides (fructose and glucose) is about 75%, and 10–15% consists of disaccharides [48,49].

The qualitative and quantitative content of sugars in honey is primarily determined by its botanical and geographical origin. However, the amount of sugars can also be slightly influenced by factors such as weather, processing, and storage conditions [11,50]. One of the important characteristics when evaluating the quality of honey is the ratio of fructose and glucose, as well as glucose and water. When the fructose-to-glucose ratio is less than 1.0, crystallization occurs more quickly, and it decreases when the ratio is greater than 1.0 [51–53].

When confirming the naturalness of honey, the concentration of sucrose is also important because a higher-than-permissible content serves as an indicator of adulteration [54].

The high-pressure liquid chromatography method was used to identify and quantify the individual components of sugars in Jara honey. Based on standard calibration, the quantitative results for each honey sample are shown in Figure 3. Among the compounds identified (fructose, glucose, sucrose, and maltose), the dominant compounds were fructose and glucose. Based on standard calibration, the quantitative results for each honey sample are shown in Table 3.

Comparing the obtained results, the concentration of fructose was 44.5–56.4% and that of glucose was 27.18–37.8% (Table 3). In particular, the fructose content of Keda's honey samples ranged from 45.175% to 52.58%, with an average of 48.79%. Glucose concentrations ranged from 37.8% to 34.97%, with an average of 32.67% glucose. In the samples from Shuakhevi and Khulo, the fructose content was higher compared to the honey from Keda,

at about 51% (average value); the average glucose concentration in the Shuakhevi honey was 31.32%, while it was 32.21% in the Khulo samples.

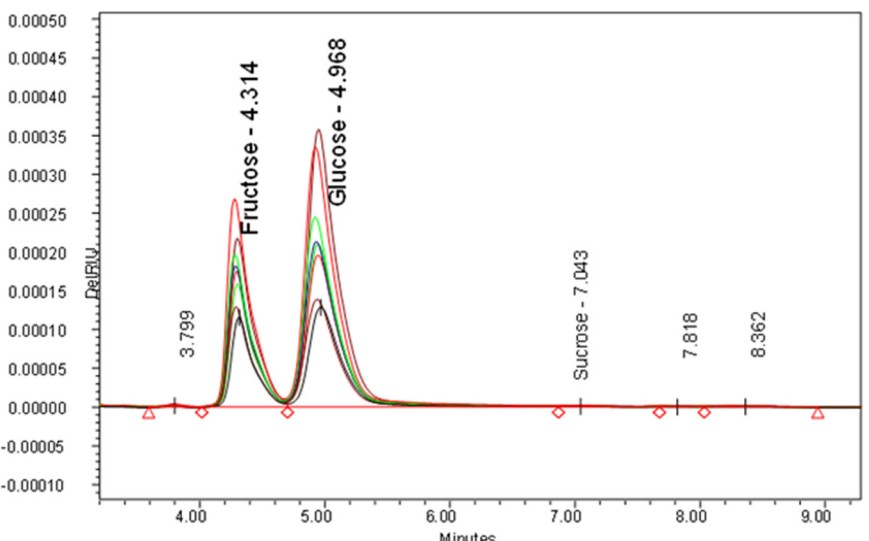

**Figure 3.** Chromatograms of the sugars in the Jara honey samples.

**Table 3.** Quality and quantity of Jara honey carbohydrates in Western Georgia.

| Samples | Fructose % | Glucose % | Sucrose % | Maltose % | The Total Amount of Carbohydrates % |
|---------|-----------|-----------|-----------|-----------|-------------------------------------|
| JH-1 | 45.175 ± 0.68 | 17.8 ± 0.27 | - | - | 82.246 ± 1.23 |
| JH-2 | 46.523 ± 0.70 | 34.974 ± 0.52 | 0.326 ± 0.0049 | - | 81.823 ± 1.23 |
| JH-3 | 48.576 ± 0.73 | 33.605 ± 0.44 | 0.065 ± 0.0010 | - | 82.246 ± 1.23 |
| JH-4 | 53.266 ± 0.80 | 29.045 ± 0.55 | - | -- | 82.312 ± 1.23 |
| JH-5 | 44.543 ± 0.67 | 36.472 ± 0.44 | - | - | 81.015 ± 1.22 |
| JH-6 | 50.998 ± 0.76 | 29.607 ± 0.43 | - | - | 80.605 ± 1.21 |
| JH-7 | 52.580 ± 0.79 | 28.590 ± 0.52 | - | - | 81.170 ± 1.22 |
| JH-8 | 46.144 ± 0.69 | 34.619 ± 0.44 | - | - | 80.763 ± 1.21 |
| JH-9 | 51.368 ± 0.77 | 29.356 ± 0.48 | - | - | 80.724 ± 1.21 |
| JH-10 | 49.541 ± 0.74 | 31.786 ± 0.43 | 0.123 ± 0.0018 | - | 81.450 ± 1.22 |
| JH-10 | 54.438 ± 0.82 | 28.651 ± 0.41 | - | - | 83.089 ± 1.25 |
| JH-12 | 54.402 ± 0.82 | 27.656 ± 0.56 | - | 2.006 ± 0.0301 | 84.064 ± 1.26 |
| JH-13 | 45.740 ± 0.69 | 37.226 ± 0.46 | 0.217 ± 0.0033 | - | 83.183 ± 1.25 |
| JH-14 | 54.457 ± 0.82 | 30.986 ± 0.51 | - | - | 85.443 ± 1.28 |
| JH-15 | 48.355 ± 0.73 | 34.036 ± 0.50 | - | 1.800 ± 0.0270 | 84.191 ± 1.26 |
| JH-16 | 49.688 ± 0.75 | 33.125 ± 0.41 | 0.403 ± 0.0060 | | 83.216 ± 1.25 |
| JH-17 | 56.400 ± 0.85 | 27.180 ± 0.54 | 0.754 ± 0.0113 | | 84.334 ± 1.27 |
| JH-18 | 46.043 ± 0.69 | 35.739 ± 0.27 | - | - | 81.782 ± 1.23 |

In the samples taken for analysis, considering the average value, fructose was the dominant sugar. (The mass share of fructose in the total sugars (average of 23 indicators) was as follows: in the Keda samples—52.58%; in the Shuakhevi samples—60.37%; in the Khulo samples—61.42%). This was followed by glucose: in the Keda samples—37.06%; in

the 23 samples from Shuakhevi—37.64%; in the samples form Khulo—38.33% (23% of the mass share of the total sugars average rate). (Table 3).

Sucrose was identified in the following samples: JH-2, JH-3, JH-10, JH-13, JH-16, and JH-17; and its content in the honey samples ranged from 0.065% to 0.403%. Maltose was identified only in two samples: JH-12 and JH-15. The maltose content was 2.006% and 1.8%. The sucrose and maltose contents were significantly lower than the fructose content as well as the glucose content (Figure 3). According to international standards, the sucrose content should not be higher than 5% [55,56].

Carbohydrates, especially glucose and fructose, are the main components of honey and are essential for crystallization. Glucose is considered the crystallizing sugar because of its reduced solubility. Certain monofloral honeys (such as citrus) naturally crystallize neatly and uniformly, but most commercial honey is considered to be of low quality.

The G/F ratios of the Jara honey samples in the present study varied between 1.2 and 2.08. The glucose/fructose ratio in the Jara honey samples was greater than 1, and the average value was 1.6 in the Keda samples, 1.63 in the Shuakhevi samples, and 1.58 in the Khulo samples (Table 4).

**Table 4.** Fructose/glucose and glucose/water ratios in Jara honey.

| Samples | Fructose/Glucose (F/G) | Glucose/Water (G/W) |
|---------|------------------------|---------------------|
| JH-1 | $1.22 \pm 0.018$ | $0.458 \pm 0.0069$ |
| JH-2 | $1.33 \pm 0.020$ | $1.96 \pm 0.0294$ |
| JH-3 | $1.45 \pm 0.022$ | $2.0 \pm 0.0300$ |
| JH-4 | $1.83 \pm 0.027$ | $2.25 \pm 0.0338$ |
| JH-5 | $1.22 \pm 0.018$ | $0.504 \pm 0.0076$ |
| JH-6 | $1.72 \pm 0.026$ | $1.63 \pm 0.0245$ |
| JH-7 | $1.84 \pm 0.028$ | $1.62 \pm 0.0243$ |
| JH-8 | $1.33 \pm 0.020$ | $1.91 \pm 0.0287$ |
| JH-9 | $1.75 \pm 0.026$ | $1.58 \pm 0.0237$ |
| JH-10 | $1.56 \pm 0.023$ | $1.75 \pm 0.0263$ |
| JH-10 | $1.90 \pm 0.029$ | $1.78 \pm 0.0267$ |
| JH-12 | $1.97 \pm 0.030$ | $1.84 \pm 0.0276$ |
| JH-13 | $1.23 \pm 0.018$ | $0.445 \pm 0.0067$ |
| JH-14 | $1.76 \pm 0.026$ | $2.16 \pm 0.0324$ |
| JH-15 | $1.42 \pm 0.021$ | $2.21 \pm 0.0332$ |
| JH-16 | $1.50 \pm 0.023$ | $2.02 \pm 0.0303$ |
| JH-17 | $2.08 \pm 0.031$ | $1.47 \pm 0.0221$ |
| JH-18 | $1.09 \pm 0.016$ | $0.5 \pm 0.0075$ |

As for the glucose/water ratio in the honey samples, it ranged from 158 to 2.22 in the Keda samples, from 1.75 to 2.24 in the Shuakhevi samples, and from 176 to 2.21 in the Khulo samples. Jara honey is characterized by a high ratio of F/G and G/W (more than 1.2) (Table 4).

### 3.3. Free Acidity meq/kg

The acid content of honey is relatively low, but it is crucial for the taste of honey. The acidity of honey is determined by titration and is expressed in milliequivalents per kg [57].

Less than 0.5% of the solids in honey are organic acids; however, these acids have a significant impact on flavor [16], since free acidity is a key indicator of the microbial deterioration of honey [58]. According to the EU Regulation (Council EU, 2001) and Codex

Alimentarius (2001), the maximum amount of free acidity that can be present in honey is 50 meq/kg [44]. The results of the examination of free acidity are shown in Table 2. The free acidity content ranged from 20.96 to 28.5 meq/kg in the Jara honey varieties, with a mean value of free acidity lower than the permitted threshold.

From the analysis results, the average acidity rate in the Jara honey ridge samples was 24.23 units, while the average rates in the Shuakhevi and Khulo samples were similar to each other at 26.06 and 26.08 meg/kg (Table 2).

Hence, the low free acid values obtained in the current work are a good indicator of conservation. These results showed that there was no unwanted fermentation.

### 3.4. pH

The pH is one of the most important characteristics of honey [59], as it may influence honey texture, stability, and shelf life [33]. In particular, it prevents the development of microbiological processes. [56].

The pH values between 3.4 and 6.1 indicate the freshness of the honey samples [60]. All of the investigated Jara honey samples were acidic and were within the limit (pH 4.23 to 5.63) (Table 2) and within the standard limit (pH 3.40–6.10) [61], ensuring the freshness of the honey samples. In particular, the average value of the pH in the honey ridge samples is 5.214, in the Shuakhevi samples—4.96, and in Khulo samples—5.02.

### 3.5. Electrical Conductivity (ms/cm)

Electrical conductivity is a very important property of honey [62]. It is greatly influenced by the concentration of organic acids and proteins, as well as by the ash content and active acidity [48]. Electrical conductivity generally falls within the range of 0.39–0.76 ms/cm [63]. Accordingly, the electrical conductivity index, along with other parameters, serves as the main marker for confirming the botanical origin of honey [64].

The bright color of honey usually points to a lower conductivity compared to dark-colored honey [65].

The values of electrical conductivity in the investigated honey samples ranged from 1.071 to 1.706 μs/sm (Table 2). The EC values in the investigated honey samples were not within the recommended range (below 0.8 ms/cm) [48].

The conductivity values of the Jara honey ridge samples range from 1.071 to 1.706, with the highest values observed in four samples: JH 5-1.706, JH 6-1.552, JH 7-1.606, and JH 9-1.447 μs/cm. In the Shuakhevi samples, the electrical conductivity ranged from 1.1008 to 1.3119 μs/cm, while in the Khulo samples, the indicator was almost the same, ranging from 1.122 to 1.66.

### 3.6. Cations

Honey contains a variety of macro and micro minerals that are minor constituents of honey, presented in the range of 0.02–1.03% [66]. These elements mainly include K, Na, Mg, Ca, P, Mn, Fe, Li, Co, etc. [11,22]. The ash content of honey is the principal source of trace elements [67].

The qualitative and quantitative content of ash is an important feature of honey, influenced by both botanical and geographical origin [11,67]. A high degree of ash content in honey is a confirmation of a high concentration of pollen. [68,69]. Minerals affect the color of honey and are found in greater quantities in dark honey compared to light honey [58].

Also, it is possible to detect falsification based on ash content. When bees are fed sugar syrup, the ash content is also low [58].

In the Jara honey, there were identified microelements such as Li, Na, K, Mg, and Ca (Figure 4), with concentrations ranging from 2613.5 to 5568.4 ppm. The analysis of the results is presented in Table 5. Based on the obtained results, it can be concluded that K is the dominant element, with its content ranging from 2174.86 to 5074.36 ppm. This mineral is the most quantitatively important in Jara honey, accounting for around 89% of the total mineral content. The average potassium ion concentrations in the honey samples from

Keda (3546.72 ppm) and Shuakhevi (3501.46 ppm) are similar. However, the concentration of potassium ions in the ridge samples varies significantly. For instance, in the first, second, third, and sixth samples, the concentration ranges from 2239.88 ppm to 2998.04 ppm, while in the fourth, fifth, and eighth honey, it ranges from 4702.58 ppm to 48603 ppm, with a difference of up to 36 ppm. In the Shuakhevi honey samples, there is no significant variation in the indicators (3259.64 ppm to 3856.74 ppm). Similar to the ridge samples, the potassium content in the Khulo honey varies. For instance, JH 16 (2356.68 ppm) and JH 18 (2395.72 ppm) have almost identical concentrations. However, compared to these samples, the potassium content is twice as high in the fourteenth sample of Khulo honey, reaching 5074.36 ppm (Table 5).

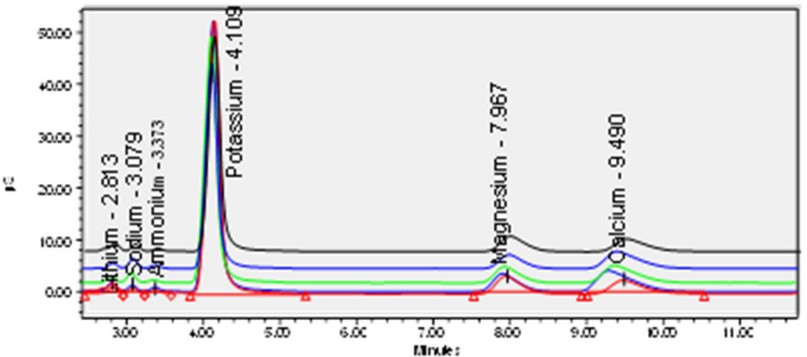

**Figure 4.** Chromatograms of the cations in the Jara honey samples.

**Table 5.** The cations in Jara honey from Western Georgia.

| Samples | Lithium ppm | Sodium ppm | Potassium ppm | Magnesium ppm | Calcium ppm | Total Ions ppm |
|---|---|---|---|---|---|---|
| JH-1 | 10.54 ± 0.26 | 17.58 ± 0.44 | 2998.04 ± 89.9 | 80.22 ± 2.4 | 270.44 ± 8.1 | 3376.8 ± 101.3 |
| JH-2 | 11.22 ± 0.28 | 17.82 ± 0.45 | 2252.34 ± 67.6 | 56.58 ± 1.7 | 292.64 ± 8.8 | 2630.6 ± 78.9 |
| JH-3 | 12.7 ± 0.32 | 18.44 ± 0.46 | 2850.78 ± 85.5 | 72.54 ± 2.2 | 289.68 ± 8.7 | 3244.1 ± 97.3 |
| JH-4 | 15.06 ± 0.38 | 10.06 ± 0.25 | 4810.24 ± 144.3 | 78.52 ± 2.4 | 376.72 ± 11.3 | 5290.6 ± 158.7 |
| JH-5 | 14.76 ± 0.37 | 2.4 ± 0.06 | 4860.36 ± 145.8 | 171.04 ± 5.1 | 245.74 ± 7.4 | 5294.3 ± 158.8 |
| JH-6 | 10.78 ± 0.27 | 29.72 ± 0.74 | 2239.88 ± 67.2 | 88.32 ± 2.6 | 373.12 ± 11.2 | 2741.8 ± 82.3 |
| JH-7 | 14.94 ± 0.37 | 2.66 ± 0.07 | 3262.64 ± 97.9 | 262.34 ± 7.9 | 795.06 ± 23.9 | 4337.6 ± 130.1 |
| JH-8 | 15.94 ± 0.40 | 4.74 ± 0.12 | 4702.58 ± 141.1 | 55.48 ± 1.7 | 181.74 ± 5.5 | 4960.4 ± 148.8 |
| JH-9 | 15.38 ± 0.38 | 6.7 ± 0.17 | 3943.58 ± 118.3 | 59.22 ± 1.8 | 141.28 ± 4.2 | 4166.1 ± 125. 0 |
| JH-10 | 15.76 ± 0.39 | 6.82 ± 0.17 | 3856.74 ± 115.7 | 79.14 ± 2.4 | 258.3 ± 7.7 | 4216.7 ± 126.5 |
| JH-10 | 10.76 ± 0.27 | 10.72 ± 0.27 | 3594.96 ± 107.8 | 41.6 ± 1.2 | 390.38 ± 11.7 | 4048.4 ± 121.5 |
| JH-12 | 17.82 ± 0.45 | 2.94 ± 0.07 | 3259.64 ± 97.8 | 85.72 ± 2.6 | 402.72 ± 12.1 | 3768.8 ± 113.1 |
| JH-13 | 12.4 ± 0.31 | 2.82 ± 0.07 | 3296.6 ± 98.9 | 83.32 ± 2.5 | 201.86 ± 6.1 | 3597 ± 107.9 |
| JH-14 | 16.48 ± 0.41 | 23.06 ± 0.58 | 5074.36 ± 152.2 | 84.12 ± 2.5 | 370.42 ± 11.1 | 5568.4 ± 167.1 |
| JH-15 | 8.22 ± 0.21 | 13.64 ± 0.34 | 3803.18 ± 114.1 | 43.2 ± 1.3 | 272.3 ± 8.2 | 4140.5 ± 124.2 |
| JH-16 | 9.84 ± 0.25 | 15.8 ± 0.40 | 2356.68 ± 70.7 | 58.5 ± 1.8 | 288.66 ± 8.7 | 2729.4 ± 81.9 |
| JH-17 | 14.74 ± 0.37 | 33.58 ± 0.84 | 2174.86 ± 65.2 | 84.02 ± 2.5 | 306.32 ± 9.2 | 2613.5 ± 78.4 |
| JH-18 | 19.44 ± 0.49 | 7.18 ± 0.18 | 2395.72 ± 71.9 | 67.42 ± 2.0 | 183.62 ± 5.5 | 2673.3 ± 80.2 |

The second most abundant mineral in all samples was Ca, ranging from 141.28–795.06 ppm. The average content of Mg and Na varied significantly among samples. The concentrations

found in the samples ranged from 41.6 to 262.34 ppm for Mg and from 2.4 to 33.58 ppm for Na. The concentration of Li was the lowest (8.22–19.44 ppm) (Table 5).

### 3.7. Color Intensity

Honey color is an important sensory characteristic ranging from colorless to dark brown [70]. The honey color completely depends on the honey plants from which the nectar is taken. The color range can also vary depending on geographical origin [15,70]. There are often cases when there is a difference in the order of honey color in the international market; for example, in the European market, there is a higher demand for honey with a dark color and strong aroma, whereas in America, there is a preference for light-colored honey with a light aroma [35]. The color intensity of honey also determines the concentration of biologically active secondary metabolites in nectar [71], and it is directly correlated with the antioxidant activity of honey [72].

Only one of the possible seven color classes (Pfund scale) was found in the Jara honeys (Table 6): dark amber. The color of the samples taken for analysis was in the range of 122.44–294.81 mm; according to the scale, it exceeds 144 mm. That indicates it is a dark amber.

**Table 6.** Total phenols. phenolic acids, flavonoids, color, and antioxidant activity of Jara honey.

| Samples | Total Phenols mg/kg | Total Phenolic Acids mg/kg | Total Flavonoids mg/kg | Antioxidant Activity DPPH-50% Inhibition mg of Samples | Color (mmPfund) |
|---|---|---|---|---|---|
| JH-1 | 801.61 ± 20.04 | 453.3 ± 11.33 | 231.8 ± 5.80 | 90.01 ± 2.25 | 181.61 ± 4.54 |
| JH-2 | 1105.56 ± 27.64 | 682.4 ± 17.06 | 321.4 ± 8.04 | 75.1 ± 1.88 | 294.81 ± 7.37 |
| JH-3 | 987.19 ± 24.68 | 543.8 ± 13.60 | 307.9 ± 7.70 | 88.29 ± 2.21 | 249.31 ± 6.23 |
| JH-4 | 872.48 ± 21.81 | 418.5 ± 10.46 | 265.8 ± 6.65 | 79.59 ± 1.99 | 189.98 ± 4.75 |
| JH-5 | 752.99 ± 18.82 | 452.1 ± 11.30 | 229.4 ± 5.74 | 94.75 ± 2.37 | 180.76 ± 4.52 |
| JH-6 | 670.61 ± 16.77 | 379.9 ± 9.50 | 203.7 ± 5.09 | 107.15 ± 2.68 | 146.68 ± 3.67 |
| JH-7 | 622.34 ± 15.56 | 388.7 ± 9.72 | 154 ± 3.85 | 128.08 ± 3.20 | 148.26 ± 3.71 |
| JH-8 | 634.22 ± 15.86 | 359.8 ± 9.00 | 185.7 ± 4.64 | 125.77 ± 3.14 | 131.11 ± 3.28 |
| JH-9 | 647.77 ± 16.19 | 419 ± 10.48 | 143.6 ± 3.59 | 111.58 ± 2.79 | 130.44 ± 3.26 |
| JH-10 | 900.55 ± 22.51 | 533.8 ± 13.35 | 291.4 ± 7.29 | 82.38 ± 2.06 | 219.16 ± 5.48 |
| JH-11 | 988.79 ± 24.72 | 580.2 ± 14. 51 | 299.7 ± 7.49 | 89.83 ± 2.25 | 271.29 ± 6.78 |
| JH-12 | 902.03 ± 22.55 | 567.2 ± 14.18 | 232 ± 5.80 | 85.88 ± 2.15 | 232.87 ± 5.82 |
| JH-13 | 931.49 ± 23.29 | 612.4 ± 15.31 | 221 ± 5.53 | 86.43 ± 2.16 | 274.29 ± 6.86 |

### 3.8. Phenolic Compounds

Phenolic compounds are one of the most significant chemicals in honey [20]. They are powerful natural antioxidants that are biologically active secondary metabolites from plants that operate at the molecular level [73]. Phenolic compounds, as secondary plant metabolites and natural antioxidants, significantly determine the biological activity of honey [20,43]. In particular, these compounds are responsible for the antioxidant activity of honey [43,74], as they can bind or neutralize free radicals [11]. The content of phenolic compounds, both qualitatively and quantitatively, depends entirely on the type of honey plants [11,35,75].

The main indicators of honey's botanical origin are polyphenols, which also have a strong medicinal and dietary value [14,76]. Honey is utilized as an essential source of phenolic compounds in the human diet due to its abundance of phenolic compounds [20,77].

The total amount of phenols in the honey samples taken for analysis ranged from 622.34 to 1105.56 mg/kg. According to the phenol content, the samples from Keda

(788.31 mg/kg) and Khulo (817.51 mg/kg) had similar phenol levels. However, in the Keda honey, there was a noticeable difference between the samples. In particular, the phenol content, including JH 6–9, averaged 643.73 mg/kg, with a relatively high content observed in the third sample (987.19 mg/kg) and the second sample (1105.56 mg/kg). A similar difference was observed in the samples of the Khulo Jara honey. The average total phenol content in the Shuakhevi honey was relatively high at 930.72 mg/kg (Table 6).

### 3.9. Flavonoids

Flavonoids are low-molecular-weight phenolic compounds based on the flavan nucleus [78], responsible for the aroma and antioxidant potential of honey [46]. Their biological effects span a wide range, including antibacterial, anti-inflammatory, anti-allergic, and antithrombotic actions [76]. According to the floral and geographic sources of the honey, flavonoid profiles typically vary greatly [35,78,79]. The following Jara honey samples had the highest flavonoid content: JH-2 (321.9 mg/kg), JH-3 (307.9 mg/kg), JH-11 (299.7 mg/kg), and JH-10 (291.4 mg/kg). They were relatively lower in JH-18 (267.1 mg/kg), JH-4 (265.8 mg/kg), and JH-17 (255.3 mg/kg) (Table 6).

### 3.10. Phenolic Acids (Aromatic Carbonic Acids)

Phenolic acids (aromatic carbonic acids) are a subclass of the most numerous and ubiquitous groups of secondary plant metabolites [78]. Phenolic acids are classified as derivatives of cinnamic and benzoic acids [70]. Their chemical structure is simple C6-C [55]. Honey contains a wide range of phenolic acids [26]. Phenolic acids are not only present in honey, but they also might distinguish some kinds of honey [78,80].

The phenolic acid content in the Jara honey samples ranged from 359.8 to 682.4 mg/kg and accounted for around 45–65% of the total phenolic content. The amount of phenolic acids (according to the average indicator) was much higher in the Shuakhevi samples, amounting to 573.40 units, while it was almost equal in the Keda (455.25 mg/kg) and Khulo (488.42 mg/kg) samples, similar to the content of the total phenols (Table 6).

The phenolic acid content was highest in the following honey samples: JH-2 (682.4 mg/kg), JH-13 (612.4 mg/kg), and JH-14 (648 mg/kg), whereas the sample JH-1 had the lowest content (453.3 mg/kg) (Table 6).

### 3.11. Antioxidant Activity by the DPPH Method

Honey is a well-known abundant source of both enzymatic (glucose oxidase and catalase) and non-enzymatic (L-ascorbic acid, flavonoids, and phenolic acids) antioxidants, which have been shown to have health-promoting anti-oxidative properties [10]. Consuming honey is a successful strategy for boosting total plasma antioxidants and reducing capacity in people [20,81]. The determination of antioxidant activity in food products by the free radical—DPPH method is a highly adapted method. The method is simple and completed in a short time [82]. 2,2-Diphenyl-1-picrylhydrazyl is a stable compound at room temperature and rapidly recovers in solution in the presence of antioxidants. The violet color of the radical disappears or turns yellow, so the absorption index also decreases at 517 nm [83]. The antioxidant activity of the analyzed sample is calculated by the difference in absorbance values. In particular, the lower the difference index, the higher the antioxidant activity of the analyzed sample. The smaller mass sample achieves 50% inhibition of the DPPH radical [84]. This method is also actively used to determine the antioxidant activity of honey because honey is rich in secondary metabolites—phenolic compounds [3,56].

Among Jara's honey samples, those with a total phenol content of 801.61 to 1105.56 mg/kg had a relatively high antioxidant activity, so less of the honey mass was required to inhibit 50% of the radical, namely 75.1 to 90, 68 mg, while 622.34–752.95 mg/kg of the total phenols were inhibited by 94.75–128.08 mg of honey. In the presented samples, the relatively high activity (75.1 mg, 50% inhibition of honey 0.1 mm DPPH) of one particular sample stood out, the honey grown in the Keda municipality—JH-2, which contained a large amount

of total phenols (1105.56 mg/kg), total phenolic acids (682.4 mg/kg), and total flavonoids (321.4 mg/kg) (Table 6).

A decrease in the amount of phenolic compounds (total phenols, total phenolic acids, and total flavonoids) in honey also leads to a decrease in antioxidant activity. The amount of phenolic compounds in Jara honey sample 7 was the lowest (total phenols 622.34 mg/kg, total phenolic acids 388.7 mg/kg, and total flavonoids 151.0 mg/kg), and more honey was required for 50% inhibition of 0.1 mm DPPH (128.08 mg) (Table 6).

A directly proportional relationship was established between the quantitative content of phenolic compounds, color, and the antioxidant activity of honey (Table 2).

*3.12. Proline*

The amino acid composition of honey is completely dependent on the botanical origin of the honey [43], and therefore its qualitative and quantitative content is successfully used as an indicator of the naturalness and quality of honey. The mass share of amino acids in honey is about 1%. Its composition includes glutamic acid, aspartic acid, glycine, threonine, histidine, glutamine, proline, and others. But among them, there is more proline, which is mainly formed when the nectar is processed. Its content depends on the time of nectar processing by the bees and, accordingly, on the origin of the honey. It is about 50–85% of the total mass of the amino acids, and its concentration is different in different honeys [43]. The regulation defines the content of proline in honey, and it should not be less than 180 mg/kg [34]. The proline content of the Jara honey samples is presented in Table 7.

**Table 7.** Proline content, diastase activity, and protein content of Jara honey.

| Samples | Proline mg/kg | Diastase Activity (Shade) | Protein % |
|---------|--------------|----------------------------|-----------|
| JH-1 | 913.98 ± 27.41 | 14.01 ± 0.21 | 0.31 ± 0.0047 |
| JH-2 | 761.28 ± 22.83 | 9.23 ± 0.14 | 0.86 ± 0.0129 |
| JH-3 | 790.62 ± 23.71 | 20.0 ± 0.30 | 0.91 ± 0.0137 |
| JH-4 | 1109.17 ± 33.27 | 16.5 ± 0.25 | 0.91 ± 0.0137 |
| JH-5 | 1248.39 ± 37.45 | 18.75 ± 0.28 | 0.43 ± 0.0065 |
| JH-6 | 1054.08 ± 31.62 | 17 ± 0.26 | 0.46 ± 0.0069 |
| JH-7 | 1311.06 ± 39.33 | 13.33 ± 0.20 | 0.46 ± 0.0069 |
| JH-8 | 1090.02 ± 32.70 | 11.56 ± 0.17 | 0.46 ± 0.0069 |
| JH-9 | 896.04 ± 26.88 | 9.68 ± 0.15 | 0.40 ± 0.0060 |
| JH-10 | 985.25 ± 29.55 | 21 ± 0.32 | 0.89 ± 0.0134 |
| JH-10 | 927.39 ± 27.82 | 13.63 ± 0.20 | 0.41 ± 0.0062 |
| JH-12 | 1234.90 ± 37.04 | 21.11 ± 0.32 | 0.43 ± 0.0065 |
| JH-13 | 1198.16 ± 35.94 | 27 ± 0.41 | 0.42 ± 0.0063 |
| JH-14 | 920.53 ± 27.61 | 21.6 ± 0.32 | 0.91 ± 0.0137 |
| JH-15 | 1097.91 ± 32.93 | 16.2 ± 0.24 | 0.42 ± 0.0063 |
| JH-16 | 1148.98 ± 34.46 | 12.9 ± 0.19 | 0.43 ± 0.0065 |
| JH-17 | 1077.14 ± 32.31 | 14.2 ± 0.21 | 0.47 ± 0.0071 |
| JH-18 | 1372.29 ± 41.16 | 18.3 ± 0.27 | 0.41 ± 0.0062 |

The proline content of the Jara honey samples ranged from 761.28 to 1372.29 mg/kg (Table 7). According to the average values of the obtained results, the proline content was similar across all samples: in the Keda samples, it was 1019.40 mg/kg, in the Shuakhevi honey, it was 1086.43 mg/kg, and in the Khulo samples, it was 1123.37 mg/kg. The lowest proline concentration was measured in JH-2 (761.28 mg/kg) and JH-3 (790.62 mg/kg) in

the honey samples from Keda. In the other honey, the proline content was higher than 800 mg/kg. The highest proline concentration was in JH-18 (1372.9 mg/kg). The highest amount was observed in five samples: JH-5 (12,498.39 mg/kg), JH-7 (1311.06 mg/kg), JH-12 (1234.9 mg/kg), JH-13 (1208.16 mg/kg), and JH-18 (1372.29 mg/kg). Table 7 shows that all the proline values for honey were well above the 180 mg of proline per kilo of honey standard.

### 3.13. The Activities of Enzymes

The activities of enzymes are the basis for evaluating the quality of honey [85]. The enzymatic composition of honey includes glucosidases, α and β amylases, α and β glucosidases, as well as proteases. Honey types differ from each other in the composition and quantity of enzymes, as their content is completely dependent on both the nectar collection period and the physiological age of the bee colony.

α and β amylases, or diastases, are the enzymes that occur in relatively large quantities in honey, and their content depends on their botanical and geographical origin. [85]. A diastase catalyzes the breakdown of starch into maltose [33,34,41].

Diastases are sensitive to heating, and their activity decreases at high temperatures. Therefore, its value is used as a marker of age and an indicator of high-temperature treatment in storage conditions [33].

Diastase activity is a honey quality parameter used to determine if the honey has been extensively heated during processing [84,86,87]. According to the Honey Quality and International Regulatory Standards, the diastase activity must not be less than or equal to 8 [61].

Active enzymes are very sensitive to high temperatures and will lose their activity when they exceed a certain temperature [33,34].

In all 18 samples of the Jara honey samples, the characteristic diastase activity was much higher than 8, ranging from 9.56 to 27.0 (Table 7).

### 3.14. Protein

Proteins are one of the main constituents that perform critical functions in food systems [88], and therefore they are the most important marker for confirming the origin and naturalness of honey [89].

Protein in honey can come from either plants or animals. Animal protein is produced by the bee itself and is composed of salivary gland secretions as well as by-products gathered during nectar collection or honey maturation, whereas plant proteins are derived from nectar and pollen gathered in the field [16]. Floral honey has a protein value between 1.0 and 1.5%, whereas honeydew honey has a protein content of around 3.0% [90]. The nitrogen content of the Jara honey samples ranged from 0.31 to 0.91% (Table 7).

### 3.15. Melissopalynology, or Pollen Analysis

Honey color, aroma, taste, and therapeutic–prophylactic properties depend on the flower's nectar, and the composition of the latter depends on those entomophilic plants that blossom during the period of honey collection [91]. The biologically active compounds of the plant (flower) are present first in nectar and then in honey, which determine flower biological activity [63,79,92].

Several markers are used to determine the botanical origin of honey, among which it is important to determine the morphological structure and concentration of honey pollen. If the honey is obtained entirely or mainly from the nectar of a specific plant or plant flower, it must have physicochemical, organoleptic, and morphological characteristics that are characteristic only of honey obtained from the nectar of a specific plant (monofloral honey) or plant flower (polyfloral honey) [93].

Honey consists of pollen grains collected by honeybees; hence, pollen taxonomy is the prerequisite to compare the pollen present in honey samples with special reference to

melissopalynological analysis. The taste, smell, and color of honey change according to the nectar of the flowers [94].

The flower origin of the Jara honey was determined by melissopalynological analysis. Pollen types were identified by comparison with reference slides of pollen collected directly from the plants in the study and reference images of pollen and apicultural plants in the literature [95,96].

Pollen types were identified by comparison to reference slides of pollen. Pollen grains were identified and quantified by applying microscopy to preparations taken from the honey samples. About 500 pollen grains were counted from each sample. All measurements were repeated to ensure significant precision.

The proportion of each type of pollen was calculated as a percentage of total pollen. The pollen concentration in honey is regulated according to European and international standards. According to the pollen content, pollen is divided into four groups: the dominant pollen (more than 45%), secondary pollen (16–45%), significant minor pollen (3–15%), and minor pollen (less than 3%) [68,69,97]. In general, honey is considered monofloral if the dominant pollen content exceeds 45%, and if there is no dominant pollen content, the honey is considered polyfloral. However, the number of dominant pollens is different for different kinds of honey; the average rate of dominant pollens is 94.5% for chestnut honey, 38.6% for rhododendron honey, 28.1% for acacia honey, and 22.9% for lime honey, etc. [97,98].

For the identification of the Jara honey's botanic origin, beekeepers used information about Jara's location.

The results of the qualitative pollen analysis indicated the diversity of resources utilized by honeybees in the region of investigation. Pollen analysis of the Jara honey showed that out of 18 analyzed samples, the following were plants were identified: chestnut, *Tilia*, acacia, *Juglans regia*, *Prunus laurocerasus*, *Malus domestica*, *Pyrus communis* L., *Prunus domestica*, *Trifolium pratense*, *Taraxacum*, *Solidago virgaurea*, *Rubus idaeus,* rhododendron, and another pollen (Table 8).

**Table 8.** Pollen types in the Jara honey.

| Pollen Types in the Honey Samples (in Percentages) | | | |
|---|---|---|---|
| **Samples** | **Dominant Pollen (>45%)** | **Secondary Pollen (16–45%)** | **Important Minor Pollen (3–15%)** | **Minor Pollen (<3%)** |
| JH-1 | - | Chestnut—37.6, *Tilia platyphyllos*—35.7 | *Prunus domestica*—11.2, *Rubus idaeus*—12.6 | *Juglans regia, Prunus laurocerasus,* Rhododenron, and eq.t. |
| JH-2 | Chestnut—50.0 | Acacia—24.2, *Trifolium pratense*—20.0 | *Prunus laurocerasus*—5.8 | - |
| JH-3 | Chestnut—65.5 | *Trifolium pratense*—14.03 | Acacia—11.74, *Tilia platyphyllos*—6.0 | *Prunus laurocerasus, Juglans regia,* and eq.t. |
| JH-4 | Chestnut—47.5 | *Umbelliferae*—36.2 | *Tilia platyphyllos*—9.6, *Prunus laurocerasus*—6.7 | - |
| JH-5 | Chestnut—62.4 | Acacia—17.74, *Trifolium pratense*—17.03 | - | Rhododenron and eq.t. |
| JH-6 | - | Chestnut—36.2, *Tilia platyphyllos*—19.0, Acacia—17.8 | *Prunus laurocerasus*—14.0, *Prunus domestica* L.—7.2, *Rubus idaeus*—5.8 | - |
| JH-7 | Chestnut—73.36 | - | *Rubus idaeus*—10.98, *Trifolium pratense*—9.13, Acacia—4.5 | *Tilia platyphyllos, Taraxacum officinale, Solidago virgaurea,* and eq.t. |
| JH-8 | Chestnut—96.89 | - | - | *Juglans regia,* Acacia, *Solidago virgaurea,* and eq.t. |
| JH-9 | Chestnut—93.3 | - | Rhododenron—4.45 | Acacia, *Tilia platyphyllos, Trifolium pratense, Umbelliferae,* and eq.t. |
| JH-10 | Chestnut —96.6 | - | - | Acacia, *Tilia platyphyllos, Trifolium pratense,* Rhododenron |
| JH-11 | Chestnut—53.28 | Acacia—33.46 | *Umbelliferae*—8.1 | *Tilia, Trifolium pratense, Taraxacum,* Rhododenron, and eq.t. |

**Table 8.** *Cont.*

| | Pollen Types in the Honey Samples (in Percentages) | | | |
|---|---|---|---|---|
| Samples | Dominant Pollen (>45%) | Secondary Pollen (16–45%) | Important Minor Pollen (3–15%) | Minor Pollen (<3%) |
| JH-12 | Chestnut—90.0 | - | Acacia—7.11 | *Umbelliferae, Taraxacum officinale*, and eq.t. |
| JH-13 | Chestnut—63.47 | *Umbelliferae*—21.88 | Acacia—6.03, Rhododenron—4.88 | *Tilia, Taraxacum,* and eq.t. |
| JH-14 | Chestnut—50.38 | Acacia—25.0 | *Trifolium pratense*—10.2, *Tilia*—11.7 | *Alnus incana, Rubus idaeus,* Rhododenron, and eq.t. |
| JH-15 | - | Chestnut—38.28, *Trifolium pratense*—25.52 | *Malus domestica*—12.87, *Pyrus communis* L.—15.9 | *Umbelliferae,* Rhododenron, and eq.t. |
| JH-16 | Chestnut—80.47 | - | Acacia—5.65, *Umbelliferae*—4.62 Rhododenron—4.26 | *Tilia platyphyllos, Trifolium pratense, Taraxacum, Solidago virgaurea,* and eq.t. |
| JH-17 | Chestnut—46.19 | Acacia—21.8 | *Trifolium pratense*—10.3, *Rubus idaeus*—8.13, *Tilia platyphyllos*—5.57, *Juglans regia*—5.02 | Rhododenron and eq.t. |
| JH-18 | Chestnut—52.3 | *Tilia*—37.8 | *Umbelliferae*—7.6 | *Taraxacum, Solidago virgaurea,* and eq.t. |

In 15 of the honey samples, the consistent amount of chestnut pollen was more than 45%; for example, in Qeda's sample, it was 47.5–96.89%, 53.28–96.6 in Shuakhevi, and 46.19–80.47% in Khulo.

Interestingly, in three samples (JH 1, JH 6, and JH 15), chestnut pollen did not show up as dominant. However, its consistency in a secondary pollen group was higher than that of *Tilia*, acacia, and *Trifolium pratense* (Table 8).

From the melissopalynology analysis results, we can conclude that Jara's honey can be classified as multifloral honey because of its botanic origin. From the melissopalynology analysis results, we can conclude that Jara's honey can be classified as multifloral honey, but in honey samples 7, 8, 9, 10, 12, and 16, the concentration of chestnut flower pollen is high (73.36% to 96.89%) (Table 8).

Grayanotoxin III consistency enhances the treatment function of honey. It was identified in multiple samples (JH 1, JH 6, JH 9, JH 10, JH 11, JH 13, JH 14, JH 15, JH 16, and JH 17) from three different municipalities (Keda, Shuaxevi, and Khulo). The consistency of rhododendron pollen was higher in samples 9, 13, and 16. These samples belong to the important minor pollen group (JH 9—4.45%, JH 13—4.88%, and JH 9—4.26%); in other cases, their consistency is lower than 3% (Table 8).

### 3.16. The Identification of Grayanotoxin-III

For grayanotoxin-III identification, ultra-performance liquid chromatography (UPLC) mass (MS) and a photodiode array (PDA) detector were used with the Jara honey samples.

The molecular weight of GTX-III is 370 g/mol, appearing at m/z 369 in negative ion mode.

A compound (Figure 5) has a retention time of 8.359 min, m/z 369 [M−H]+, λ max 289 nm; according to the obtained results and the compound mass database METLIN (https://metlin.scripps.edu, accessed on 1 July 2024), the following apply: substance 1 is grayanotoxin-III—$C_{20}H_{36}O_6$ Negative FABMS: m/z = 369.26 [M−H+]; molecular weight: 370 g/mol.

For comparing chromatography analyses, rhododendron's flower and mad honey samples (Figure 5) were used, where the consistency of the toxin was much higher than in the Jara honey samples. An almost equal quantity of grayanotoxin-III was found in samples 9, 13, and 19 (Figure 5). It was proportional to rhododendron pollen. In other samples of Jara honey, grayanotoxin-III was left as a small portion, and the consistency of the pollen was much lower (minor pollen (<3%)) (Table 8).

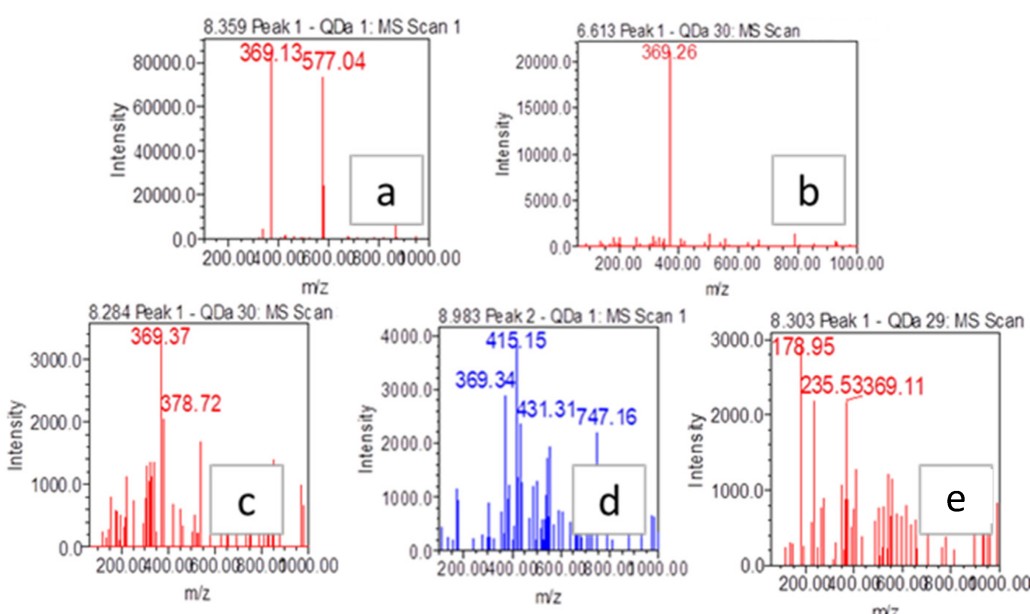

**Figure 5.** Chromatograms: (**a**) of flower rhododendrons, (**b**) of toxic honey, (**c**) of Jara honey sample 13, (**d**) of Jara honey sample 9, (**e**) of Jara honey sample 16—scan ESI-MS m/z: 369 [M−H+].

All 18 samples of Jara honey taken for analysis are dark in color, have a bitter-sweet taste, and have an extremely strong and pleasant aroma. This is the first survey about Jara honey's physicochemical and biochemical characteristics. Western Georgia's Jara honey is high in quality and rich in bioactive compounds. The physical and chemical characteristics of the honey are according to European standards. The honey samples' moisture levels vary and are compatible with honey codices. The average values between the total free acidity content, pH, and conductivity (Table 2) show a positive linear correlation. Jara's honey has the highest pH and conductivity values of 4.23 and 1.706 (μs/sm), respectively.

By comparing the result obtained in the determination of diastatic activity as one of the important characteristics (the activity index is higher than 8 units), it can be concluded that all samples of the Jara honey are natural. Dominant sugars in the form of fructose and glucose were identified in Jara honey samples by the high-pressure liquid chromatography method. A strong correlation was found between phenolic compounds, phenolic acids, and flavonoids. Color intensity increases with the increase of phenolic compounds (0.9547), phenolic acids (0.9637), and flavonoids (0.7672) (Table 2). A stable positive correlation was observed between color intensity and phenolic compounds (0.954745808) (Table 9). In addition, there is a strong correlation between antioxidant activity and biologically active compounds, so there is a linear association (Table 9).

**Table 9.** Correlation between the color intensity of honey samples and biochemical parameters.

| Correlation between the Color Intensity of Honey Samples and Biochemical Parameters | Total Phenols mg/kg | Phenolic Acids mg/kg | Total Flavonoids mg/kg | Antioxidant Activity DPPH-50% Inhibition mg of Samples | Color (mmPfund) |
|---|---|---|---|---|---|
| Total phenols mg/kg | 1.00000 | 0.92120 | 0.88781 | −0.86301 | 0.95475 |
| Phenolic acids mg/kg | 0.92120 | 1.00000 | 0.67985 | −0.71814 | 0.96380 |
| Total flavonoids mg/kg | 0.88781 | 0.67985 | 1.00000 | −0.82727 | 0.76718 |
| Antioxidant activity DPPH-50% inhibition mg of samples | −0.86301 | −0.71814 | −0.82727 | 1.00000 | −0.74267 |
| Color (mmPfund) | 0.95475 | 0.96380 | 0.76718 | −0.74267 | 1.00000 |

The obtained results showed that protein and proline contents as well as enzyme activities were within the limit that ensures the freshness of the Jara honey samples. Potassium is the most abundant element in all the samples analyzed. This mineral is the most

quantitatively important in honey, accounting for around 89% of the total mineral content. From the melissopalynology analysis result, we can conclude that Jara's honey can be classified as multifloral honey because of its botanic origin.

## 4. Conclusions

This study encompasses the characterization of semi-wild Jara honey, distinguished by its origin. The results of this study show that all honey samples indicate a good level of quality, maturity, and freshness. Physicochemical and chemical analyses of Jara honey are in agreement with European and Georgian legislation. Considering the pollen profile and physicochemical characteristics, the samples were classified as multifloral honey. Because of Grayatoxin 3's consistency, it is inevitable to take precautions when using Jara honey. The uniqueness of Jara is due to its production and harvesting methods. This study will allow us to obtain the results of the first targeted study on semi-wild Jara honey.

The designated project has been fulfilled by the financial support of the Georgia National Science Foundation (Grant AP/96/13, "Research of bioactive compounds and biological activity of pollens of honey for authentication of botanical origins of honey extracted in west Georgia", PHDF-22-3218). Any idea in this publication is possessed by the author and may not represent the opinion of the Georgia National Science Foundation.

**Author Contributions:** Methodology, I.K. and R.D.; Formal analysis, N.A. and M.K. (Meri Khakhutaishvili); Investigation, I.D. and M.K. (Maia Kharadze); Data curation, M.V. and A.K. All authors have read and agreed to the published version of the manuscript.

**Funding:** This research was funded by Shota Rustaveli National Science Foundation of Georgia grant number N 22-3218 and The APC was funded by The project Investigation of "Pollen, Bioactive Compounds, and Biological Activity of Honey to Determine the Botanical Origin of Honey Collected in Western Georgia". The Batumi Shota Rustaveli State University was also a co-financier.

**Institutional Review Board Statement:** Not applicable.

**Informed Consent Statement:** Not applicable.

**Data Availability Statement:** The original contributions presented in the study are included in the article, further inquiries can be directed to the corresponding authors.

**Conflicts of Interest:** The authors declare no conflict of interest.

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
