# Peer review of "Physicochemical Characterization and Antioxidant Activity of Jara Honey Produced in Western Georgia"

_applsci, doi:10.3390/app14166874_

Round 1

Reviewer 1 Report

Comments and Suggestions for Authors

This article summarizes the chemical composition of Jara honey, it is a very interesting subject, however, there are certain issues that need to be reexamined:

1. presentation of results - I believe that the data presented in Figures 2, 5, 8, 10, 11 should rather be shown in tables (or in one table). In Fig. 2 it is not specified in what units the free acidity is expressed, and in Fig. 8 - in what units the antioxidant activity is expressed. Please complete this. By the way, I propose that the expression antioxidant activity be replaced by the expression antiradical activity because this is what the authors tested. The title of Fig. 2 should be below the figure. Figure no. 9 is illegible and, in my opinion, does not show the relationship/correlation between individual variables. Did the authors statistically examine the occurrence of such relationship? If so, please supplement the description of methods with statistical analysis methods.

- lines 257-258, 263-264, 262-363: references needed

- line 316 - what the numbers in brackets mean (113, 16)?

2. Disscussion - this part of manuscript requires substantial revisions, or even a complete rewrite, the authors summarized the results of their research, there is no comparison with other publications, for example regarding the quality of other honeys from Georgia. This part should be called a summary rather than a discussion.

3. There are numerous formatting errors, such as:

- line 119 - space missing

- line 128 - space missing, unnecessary period,

- line 132 - unnecessary hyphen, etc.

There are many errors of this type, so I ask the authors to read the text carefully and correct it.

Reviewer 2 Report

Comments and Suggestions for Authors

The manuscript titled: "Physicochemical characterization and antioxidant activity of Jara honey produced in Western Georgia" is a very interesting source of knowledge about the composition and properties of Jara honey. The article can be published after completing several issues:

1) The authors wrote in the text of the manuscript that: “All 18 samples of Jara honey taken for the analysis are dark in color, have a bitter-sweet taste, and have an extremely strong and pleasant aroma.”

However, I missed at least a few photos of samples of this honey. I think it would significantly enrich the manuscript.

2) The authors wrote in the text of the work that: "Depending on the melissopalynology analysis result, we can conclude that Jara`s honey can be classified as multi-floral honey because of its botanic origin." and also presented Figure 11 - Pollen types in the Jara honey.

However, I missed at least a few photos of the plants from which Jara honey is obtained.

3) To emphasize more the importance of the research presented in the manuscript, it would be necessary to broadly discuss the previous uses of Jara honey (e.g. in folk medicine), the current uses of this honey in medicine, cosmetics and the food industry, and also show possible future uses of this honey (in medicine, cosmetics, food industry but also in other sectors of the economy). This can be represented by a diagram consisting of 3 main parts.

4) What is the price of Jara honey? Is it only available in Georgia? There is no information on this subject in the text of the manuscript.

5) Wax is also produced during honey production? Have the authors also studied this product?

Reviewer 3 Report

Comments and Suggestions for Authors

Dear Authors, this is an interesting and novel research on a type of honey that hasn't been adequatly explored. However, the manuscript has flaws which should be improved in order for it to be suitable for publication. I advise downloading several articles from the Applied Sciences website to see how the sections and subsections should be formed (e.g. text should be written separately from the title of the subsection). Also the references should be adjusted according to the MDPI reference guide.

The materials and methods section is very poorly described and should be revised by adding basic descriptions of procedures that would allow the readers to repeat the analysis. Also there is no description of the applied statistical analysis. 

There is not enough discussion of the results compared to other types of honey to put into context the quality of the Java honey. 

My specific comments include:

Line 10 Jhar or Java

12 no need for citation in the abstract

24 add italic for the Latin name

25-26 I advise removing this information as it is further described in detail in lines 45-46

60 The use of pesticides....? Not a full sentence

68-69 please complet the sentence

70 how did you create a product if the aim was to analyse an existing product?

76 This sentence is excess, add the information that the beekepers were private in line 73

82 cite the website according to the MDPI reference guidline

83-84 please use passive form

86-91 Information about the providers missing (e.g. Mettler Toledo, Ohio, USA)

10,104,106,108,109,112-131 information about precedures missing

132-136 LOD, LOQ, collision energy etc. please add information

Figure 2 standard deviation missing

Lines 153-154 free acids, pH and electrical conducitvity are not on this figure

Line 156 name of the Figure missing

172 which Figure, please check

Figure 4 standard deviation missing

Figure 5 is not mentioned in text

Figure 9 it is unclear from the figure what the dots are. ALso how was the correlation calculated? Should be added int omaterials and methods

Figure 10 not marked properly

Figures 12-16 excess chromatograms, could be added into supplementary material

Comments on the Quality of English Language

The formation of sentences makes the text of the manuscript hard to follow in some parts. 

Round 2

Reviewer 3 Report

Comments and Suggestions for Authors

Dear Authors, 

thank you for your replies. The revised version of the manuscript has improved compared to the first one, however there are still issues to be addressed. 

-Line 152: the number of subsection is 2.10.; line 158 2.11 and so on. 

Reply to Response 2: I am not interested in your other manuscripts. I am saying that you cannot claim that you have concluded that Jara honey is a unique product if you haven't brought it into context of other existing honeys and stated in which way do the physico-chemical properties distinguish from them, especially with regards to polyphenolic content and antioxidant activity. 

Reply to response 17: FIgures need to be able to stand on their own. There is no need for adding appendices containing the exact same information. If you chose to present the results as figures, then the figures should contain all of the necessary information, including the standard deviation. Also, appendices or supplementary material are not mentioned anywhere in the text of the manusccript.

Reply to comment 24:  The table is a clearer form. Information is missing about which of the obtained values were statistically significant (p<0.05).

-references list has not been updated according to the applied changes and it is also not uniformally shaped according to MDPI guidelines.

Comments on the Quality of English Language

Proof reading and minor editing of grammar required.

Author Response

Comments 1: Line 152: the number of subsection is 2.10.; line 158 2.11 and so on. 

Response 1: It has been corrected.

Comments 2: Reply to Response 2: I am not interested in your other manuscripts. I am saying that you cannot claim that you have concluded that Jara honey is a unique product if you haven't brought it into context of other existing honeys and stated in which way do the physico-chemical properties distinguish from them, especially with regards to polyphenolic content and antioxidant activity. 

Response 2: It is corrected.

Comments 3:  Reply to response 17: FIgures need to be able to stand on their own. If you chose to present the results as figures, then the figures should contain all of the necessary information, including the standard deviation. Also, appendices or supplementary material are not mentioned anywhere in the text of the manusccript.

Response 3: All diagrams have been changed to tables.

Comments 4: There is no need for adding appendices containing the exact same information. Reply to comment 24:  The table is a clearer form. Information is missing about which of the obtained values were statistically significant (p<0.05).

Response 4: It is corrected.

Comments 5: -references list has not been updated according to the applied changes and it is also not uniformally shaped according to MDPI guidelines.

Response 5: The references list has been corrected